# The Impact of Nonalcoholic Fatty Liver Disease on Severe Community-Acquired Pneumonia Outcomes

**DOI:** 10.3390/life13010036

**Published:** 2022-12-23

**Authors:** Branimir Gjurašin, Mia Jeličić, Marko Kutleša, Neven Papić

**Affiliations:** 1University Hospital for Infectious Diseases Zagreb, 10000 Zagreb, Croatia; 2School of Medicine, University of Zagreb, 10000 Zagreb, Croatia

**Keywords:** nonalcoholic fatty liver disease, NAFLD, community-acquired pneumonia, CAP, metabolic syndrome, acute respiratory distress syndrome, ECMO, ARDS, influenza

## Abstract

Community-acquired pneumonia (CAP) is one of the leading causes of morbidity and mortality, while nonalcoholic fatty liver disease (NAFLD) is the most common cause of chronic liver disease. NAFLD is associated with systemic changes in immune response, possibly linked to CAP severity. However, the impact of NAFLD on CAP outcomes has not been determined. The aim of this study was to evaluate clinical course, complications and outcomes of severe CAP requiring ICU treatment in patients with NAFLD in the pre-COVID-19 era. A retrospective cohort study included 138 consecutively hospitalized adult patients with severe CAP admitted to the ICU during a 4-year period: 80 patients with NAFLD and 58 controls. Patients with NAFLD more frequently presented with ARDS (68.7% vs. 43.1%), and required invasive mechanical ventilation (86.2% vs. 63.8%), respiratory ECMO (50% vs. 24.1%), and continuous renal replacement therapy (62.5% vs. 29.3%). Mortality was significantly higher in the NAFLD group (50% vs. 20.7%), and the time from hospital admission to death was significantly shorter. In survival analysis, NAFLD (HR 2.21, 95%CI 1.03–5.06) was associated with mortality independently of other components of metabolic syndrome. In conclusion, our study identified NAFLD as an independent predictor of mortality in patients with severe CAP.

## 1. Introduction

Community-acquired pneumonia (CAP) is a frequent reason for hospital admission and the most common cause of septic shock and mortality within the intensive care unit (ICU), the latter reaching up to 25–30% [1,2]. There is growing evidence that the components of metabolic syndrome are associated with higher risk of acquiring CAP or worse CAP outcomes. Metabolic syndrome, defined by a combination of abdominal obesity, dyslipidemia, arterial hypertension or insulin resistance, is becoming increasingly prevalent, with more than a 50% increase in less than a decade [3]. While each component of metabolic syndrome was reported to increase mortality of CAP, their combined impact may be even more pronounced. Patients with either insulin resistance or type 2 diabetes mellitus (T2DM) were shown to have a 1.3- and 2.8-fold increase in CAP mortality, respectively [4]. Similarly, obese patients more frequently develop CAP and require ICU treatment [5,6]. In COVID-19, dyslipidemia was associated with increased disease severity and hospital mortality [7]. 

However, the studies reported conflicting and inconsistent results [8,9,10,11,12,13,14], and did not include nonalcoholic fatty liver disease (NAFLD) as a potentially important variable. NAFLD is closely related with metabolic syndrome and is the most common chronic liver disease, affecting about 25% of the Western population [15,16]. Furthermore, NAFLD is linked to chronic low-grade inflammation and impaired immune responses with possible impact on susceptibility and outcomes of severe infections [17]. Recent insights in the “liver-lung axis” suggested the critical role of liver acute phase proteins in modulating host defense and pulmonary inflammation [18,19].

In the pre-COVID era, data associating NAFLD with adverse outcomes in serious infections were scarce, although consistent [20,21,22,23,24,25]. In COVID-19, numerous studies reported an association of NAFLD with increased COVID-19 severity and mortality [26,27,28,29].

Surprisingly, although CAP is one of the leading causes of death with a 38% increase in a 10-year period in patients with NAFLD [30], only one study examined its association with non-severe CAP outcomes [23]. Therefore, the impact of NAFLD on CAP outcomes remains unknown. Hence, the aim of this study was to analyze the clinical course, complications and outcomes of severe CAP requiring ICU treatment in patients with NAFLD. 

## 2. Materials and Methods 

### 2.1. Study Design and Patients

We conducted a retrospective cohort study that included consequently hospitalized adult patients due to severe CAP requiring ICU treatment at the University Hospital for Infectious Diseases Zagreb (UHID), Croatia, a national referral center for infectious diseases and respiratory extracorporeal membrane oxygenation (ECMO), during a 4-year period (2016 to 2019). Patients were required to have severe disease as defined by at least one of the following: Pneumonia Severity Index (PSI) > 130 (Fine class V); mechanical ventilation (invasive or noninvasive) due to acute respiratory failure; high-flow oxygen therapy with a FiO2 of 50% or more and a PaO2/FiO2 ratio < 300 mmHg; hemodynamic failure; or decreased level of consciousness. The predefined exclusion criteria were: immunosuppression; malignancies; autoimmune diseases; pregnancy; HIV; chronic viral hepatitis; presence of other chronic liver disease (hemochromatosis, Wilson’s disease, toxic hepatitis, deficiency of alpha-1-antitrypsin, or liver autoimmune disease); and consumption of alcohol > 20 g/day. Healthcare-associated and nosocomial pneumonia were not included. During the period studied, a total of 329 patients fulfilled the inclusion criteria. Next, only patients who had liver imaging data (abdominal ultrasonography) to assess liver steatosis were included (n = 196, 59.6%). Of those, a total of 58 patients were excluded: significant alcohol intake in 18, chronic viral hepatitis in 8, cirrhosis in 8, hepatotoxic medications in 9, pregnancy in 3, immunosuppression in 8, and 4 patients who had multiple ICU admissions during the same hospitalization. Finally, 138 patients were included in the study—80 with and 58 without NAFLD, as described in a flowchart, Figure 1. The study was conducted in accordance with the Declaration of Helsinki and approved by the Ethical Committee of the UHID in Zagreb, Croatia (protocol code 01-1247-2-2019, date of approval 30 August 2019).

### 2.2. Data Collection, Definitions and Outcomes

Routine demographic, clinical, microbiological, treatment and laboratory parameters were collected, including comorbidities (as measured by Charlson age–comorbidity index), chronic medications, body mass index (BMI), and nursing home residence. Selected laboratory data obtained on admission to the ICU were analyzed: C-reactive protein (CRP), procalcitonin, lactate, white-blood-cell count (WBC), neutrophil-to-lymphocyte ratio, platelet count, hemoglobin, blood urea nitrogen (BUN), serum creatinine, glucose, prothrombin time, aspartate aminotransferase (AST), alanine aminotransferase (ALT), gamma-glutamyl transferase (GGT), alkaline phosphatase (ALP), bilirubin and serum albumin concentration. Data on etiology, treatment and complications during hospitalization were collected. APRI and FIB-4 scores as surrogate markers of liver inflammation were calculated [31,32], as well as SOFA score to estimate disease severity [33]. 

Patients were diagnosed with NAFLD according to current guidelines [15,16] that require: (1) evidence of liver steatosis on ultrasonography, (2) no significant alcohol consumption, (3) no competing causes of liver steatosis, and (4) no coexisting liver disease. The liver steatosis was assessed by ultrasound in all included patients by an experienced radiologist during hospitalization, and defined as the finding of liver parenchyma with increased echogenicity and sound attenuation, as compared to kidney and spleen echogenicity [34]. Competing causes of liver steatosis and coexisting liver diseases were based on previously described exclusion criteria collected from patients’ medical charts. As per ICU protocol, all patients were tested for HIV, HBsAg and anti-HCV on ICU admission. 

Acute respiratory distress syndrome (ARDS) was classified according to the Berlin criteria [35]. Indications for respiratory veno-venous ECMO were according to ELSO guidelines, including persistent hypoxemia or hypercapnia with respiratory acidosis despite rescue strategies [36]. 

The primary outcomes measured were ICU and in-hospital mortality. Secondary outcomes included: the proportion of patients requiring invasive mechanical ventilation (IMV) and duration of IMV; severity of ARDS as defined by the Berlin criteria; the need for vv-ECMO; acute renal failure and treatment with continuous renal replacement therapy (CRRT); duration of ICU stay; and nosocomial infections.

### 2.3. Statistical Analysis

Clinical characteristics, laboratory and demographic data were evaluated and descriptively presented. The Shapiro–Wilk test was used to check if a continuous variable followed a normal distribution. Fisher’s exact test and the Mann–Whitney U test were used to compare two groups. All tests were two-tailed; a *p*-value < 0.05 was considered statistically significant. Survival analysis was evaluated using the Kaplan–Meier method, and the comparison between groups was made using the log-rank test. Risk factors associated with negative outcomes were investigated using a univariate, and subsequently a multivariable, Cox regression model by estimating the hazard ratio (HR) and its 95% confidence intervals (95%CI). Statistical analyses were performed using GraphPad Prism Software version 9.4.1. (San Diego, CA, USA).

## 3. Results 

### 3.1. Baseline Patients’ Characteristics 

Overall, 138 patients were included in the study (95; 68.84% males, a median age of 61 [IQR 49–68] years). Eighty patients (57.97%) were diagnosed with NAFLD. Patients with NAFLD more frequently had type 2 diabetes mellitus (T2DM), arterial hypertension and obesity with a higher median BMI, as presented in Table 1. There were no differences with respect to age, gender, other comorbidities and medication use, except for metformin and insulin which were more frequently prescribed in patients with NAFLD.

The time since symptom onset to hospital and ICU admission was similar between groups (5 [3–7] days vs. 4 [3–7] days; and 6 [4–8] days vs. 5 [3–8] days, respectively). Twenty-three patients were admitted after the worsening in their clinical status in the ward. 

At admission, SOFA score was high in both groups (7 [5–10] vs. 7 [4–10], *p* = 0.4121). The main reason for ICU admission was respiratory failure (65, 81.25% vs. 44, 75.86%). Thirty-four (42.5%) patients in the NAFLD group were hypotensive at admission, and eighteen (22.5%) of them were in circulatory shock. In the non-NAFLD group, 29 (50%) patients were hypotensive and 22 (37.93%) were in shock. 

Patients with NAFLD had significantly higher AST, ALT, GGT and LDH levels. There were no significant differences in CRP, procalcitonin, lactate, WBC, platelets, fibrinogen, INR, d-dimer, BUN, creatinine, total bilirubin, ALP and albumins, as shown in Table 2. While the APRI score was similar between groups, the FIB-4 score was significantly higher in patients with NAFLD (4.7 [2.1–9.3] vs. 3.1 [1.3–7], *p* = 0.0482).

No significant difference was shown in regard to etiology between the two groups, with influenza being the most common (34, 42.50% vs. 22, 37.93%). *Streptococcus pneumoniae* was the most common bacterial cause of the disease (10, 12.50% vs. 8, 13.79%). *Legionella* was diagnosed in seven patients (four in NAFLD and three in non-NAFLD group). A total of 29 (36.25%) patients in the NAFLD group and 21 (36.21%) patients in the non-NAFLD group were etiologically negative, as shown in Table 3. 

There were no differences in antibiotic treatment between groups. Before hospital admission, 38 patients received antibiotic therapy. As per the hospital ICU protocol, all patients with influenza received oseltamivir. Overall, 127 patients were treated with beta-lactams, and macrolides were initiated in a large proportion of this population (n = 93, 67.39%); 37 (26.81%) patients received fluoroquinolones; and 47 (28.98%) received corticosteroids for ARDS treatment.

### 3.2. Clinical Course and Outcomes of CAP

Patients with NAFLD more frequently presented with ARDS (55, 68.75% vs. 25, 43.10%), and more frequently developed severe ARDS, as shown in Table 4. Furthermore, patients with NAFLD more frequently required invasive mechanical ventilation (IMV), respiratory ECMO, and continuous renal replacement therapy (CRRT) during hospitalization. The duration of hospitalization was similar between groups (19 [14–35] vs. 24 [14–42], *p* = 0.1277). 

There were no differences in complications during hospitalization between the two groups, except for hospital-acquired sepsis (27, 33.75% vs. 10, 17.24%). 

Next, we performed a series of multivariable logistic regression analyses to identify risk factors associated with previously predefined outcomes of CAP. The need for invasive mechanical ventilation was associated with BUN ≥ 12 mmol/L (OR 2.31, 95%CI 1.08–6.09), LDH ≥ 600 IU/L (OR 22.88, 95%CI 2.89–89.8), and NAFLD (OR 3.31, 95%CI 1.37–8.06). Factors associated with severe ARDS were arterial hypertension (OR 2.66, 95%CI 1.17–6.06), NAFLD (OR 5.24, 95% 2.32–11.08), and influenza (OR 2.92, 95%CI 1.28–6.64). The need for ECMO was associated with a FIB-4 score ≥ 5 (OR 2.5, 95% CI 1.19–5.23), admission glucose ≥ 8.5 mmol/L (OR 3.43, 95%CI 1.55–7.60), and severe ARDS (OR 2.43, 95%CI 1.12–5.27%). Independently associated with CRRT was only ECMO treatment (OR 41, 95%CI 12.99–130.91%). Nosocomial infections were associated with ECMO treatment (OR 4.81, 95%CI 2.21–10.44) and male sex (OR 2.53, 95%CI 1.13–5.69) in our analysis.

However, mortality was significantly higher in the NAFLD group (40, 50.00% vs. 12, 20.69%, *p* = 0.0006). Time from hospital admission to death was significantly shorter in the NAFLD group (18 [9–25] vs. 46 [18–99] days, *p* = 0.0095). The median age of non-survivors was similar between groups (61 [48–69] vs. 64 [43–71] years, *p* = 0.9019), which was comparable to in survivors (60 [50–69] years). 

Early mortality, within 7 and 14 days of ICU admission, was more frequent in the NAFLD group (nine, 11.25% vs. one, 1.72% *p* = 0.0446 and 16, 20% vs. 2, 3.44%, *p* = 0.0043, respectively). Similarly, 28-day mortality (32, 40% vs. 5, 8.6%, *p* < 0.0001) and 90-day mortality (39, 48.7% vs. 9, 15.5%, *p* < 0.0001) were higher in the NAFLD group (Table 4). 

The reported cause of death in non-NAFLD patients was septic shock in eight, neurological impairment in two, and MODS in two patients. In patients with NAFLD, septic shock was reported in 18, cardiogenic shock in 11, acute myocardial infarction in 4, massive pulmonary embolism in 2, MODS in 3, and hemorrhage in 2 patients. 

### 3.3. Factors Associated with Mortality 

Next, we analyzed the risk factors associated with in-hospital mortality. In the univariable analysis, T2DM, NAFLD, a SOFA score of ≥ 8, glucose on ICU admission of ≥ 8.5 mmol/L, BUN ≥ 12 mmol/L, an LDH of ≥ 600 IU/L, d-dimer ≥ 4.0 mg/L, nosocomial infections, CRRT, severe ARDS, and ECMO were associated with mortality, as shown in Figure 2. Age, gender, other comorbidities, or laboratory findings were not associated with mortality.

However, in multivariable Cox proportional hazards regression analysis after adjustment for potential confounders, only NAFLD (HR 2.21, 95%CI 1.03–5.06), ICU admission glucose ≥ 8.5 mmol/L (HR 2.30, 95%CI 1.21–4.53), CRRT (HR 2.72, 95%CI 1.29–5.91), and severe ARDS (HR 3.24, 95%CI 1.53–6.71) remained independently associated with mortality, as shown and Figure 2 and Figure 3. 

### 3.4. NAFLD Is Associated with Increased Mortality in Patients with Influenza, Severe ARDS and Patients Requiring Respiratory ECMO 

We further analyzed the impact of NAFLD on mortality in subgroups of patients with influenza, severe ARDS and those treated with ECMO. In survival analysis, in all three groups, NAFLD was significantly associated with increased mortality as measured by log-rank test and presented with Kaplan–Meier curves in Figure 4. 

### 3.5. Impact of NAFLD and Coexisting Components of Metabolic Syndrome on Mortality 

Patients were stratified by the concomitant presence of NAFLD and other components of metabolic syndrome, T2DM and obesity. While obesity did not independently increase mortality in patients with and without NAFLD, the risk of mortality was even higher when NAFLD was combined with T2DM, as presented in Figure 5.

## 4. Discussion

In this retrospective cohort study, we examined the impact of NAFLD on severe CAP outcomes. We found a significant association of NAFLD with CAP severity in terms of a higher need for invasive mechanical ventilation, respiratory ECMO and continuous renal replacement therapy. Furthermore, patients with NAFLD had significantly higher early and late mortality, and this appears to be independent of other components of metabolic syndrome. 

To the best of our knowledge, there are only two studies that investigated the association between NAFLD and CAP. The first one showed that patients with CAP have a higher prevalence of NAFLD [37], while the other evaluated the impact of NAFLD on CAP outcomes [23]. The latter study was a retrospective cohort study that included 561 adult patients with CAP admitted to the internal medicine department [23]. This study found significant differences between the NAFLD and non-NAFLD group regarding BMI, ALT, GGT, and 30-day all-cause mortality, similar to our study. The 30-day mortality difference was 17% vs. 5.82% [23]. This study excluded patients with the most severe forms of the disease (PSI Class V), while our study included only severe CAP requiring ICU admission, which explains the higher 28-day mortality in our cohort, possibly indicating a greater impact of NAFLD on severe forms of CAP. Additionally, the authors found that patients with a calculated fibrosis score > 2 had a worse prognosis [23]. Likewise, there was a significant difference in the FIB-4 score in our cohort, suggesting that optimal liver function plays a significant role in pneumonia.

NAFLD should be viewed in the context of metabolic syndrome, specifically T2DM and obesity. Indeed, various studies have shown a correlation between a higher risk of acquiring CAP or worse CAP outcomes with the presence of T2DM [12,38,39,40]. A recent meta-analysis showed that T2DM is associated with increased 90-day post-discharge mortality, while in-hospital hyperglycemia, but not T2DM alone, is associated with increased in-hospital mortality and ICU admission [41]. Similarly, we found ICU admission glucose, not T2DM, associated with mortality in our cohort.

The studies examining the impact of obesity reported conflicting and inconsistent results; while some have identified an increased mortality risk in obese patients, including H1N1 influenza and severe COVID-19 [8,42,43,44], others have shown a mortality benefit in obese patients with CAP, known as the “obesity paradox” [9,10,11,13,14,45]. Studies reporting the obesity paradox could be confounded by selection bias and reverse causation, such as smoking, higher rates of comorbidities, weight loss due to chronic diseases, or increased infection susceptibility in obese patients, as previously reviewed [46,47]. Notably, none of the mentioned studies included NAFLD as a potentially important variable. While patients with NAFLD had a higher BMI in our cohort, obesity was not independently associated with mortality. Furthermore, obesity did not increase mortality in subgroups of patients with or without NAFLD. 

There are several theories linking NAFLD and infections. A balanced pro-inflammatory and anti-inflammatory liver response is necessary for the successful resolution of inflammation and infection. This might be distorted in NAFLD due to impaired sinusoid microcirculation resulting in aberrant hepatic microbial clearance; innate immunity and neutrophil dysfunction in the setting of insulin resistance; small bowel intestinal overgrowth; and increased intestinal permeability [48,49,50,51]. The “lung-liver axis”, a recently described bidirectional interplay model, highlights the importance of acute-phase proteins in inflammation of lung parenchyma [18,19]. Liver dysfunction affects inflammatory processes in the lung due to the impairment of reticuloendothelial activity, inadequate secretion of cytokines, and vasoactive inflammatory mediators. Meanwhile, cytokines stemming from alveolar macrophages can modify the acute-phase response of the liver [18,19]. The pathological pathways connecting NAFLD and pneumonia, however, are yet to be fully understood. 

The overall mortality in our cohort was high (37.68%), which is related to the high rate of severe ARDS and ECMO treatment due to the referral bias. As in other studies, acute kidney injury and CRRT were associated with mortality, but not with age and sex [52]. Notably, patients with NAFLD more frequently developed AKI and required CRRT, reflecting a more severe disease. Similar was described in a cohort of patients with invasive group B streptococcus infections and NAFLD [25]. 

The main limitation of this study arises from the following factors: its retrospective design; the diagnosis of NAFLD was determined by ultrasound; long-term follow-up was not undertaken; not all components of metabolic syndrome could be evaluated such as insulin resistance, dyslipidemia, or types of obesity; steatosis grade was not analyzed due to the lack of data; not all hospitalized patients were routinely screened for liver steatosis and since there was no protocol for US in ICU patients with CAP, this could lead to confounding by indication bias; and UHID is a national referral center for respiratory ECMO with a higher percentage of the most severe CAP admissions which might result in referral bias. Nevertheless, we report a well-defined cohort of patients and provide the first evidence of NAFLD impact on severe CAP outcomes. 

## 5. Conclusions

In conclusion, in this retrospective cohort study, we report a significant impact of NAFLD on CAP severity and outcomes. NAFLD was associated with increased mortality, irrespective of other components of metabolic syndrome. Furthermore, this was consistent in all patients’ subgroups, including influenza, severe ARDS and respiratory ECMO. Our data contribute to the growing evidence describing the role of NAFLD in infections and highlights the importance of including NAFLD as a variable in future studies investigating CAP outcomes and treatment strategies.

## Figures and Tables

**Figure 1 life-13-00036-f001:**
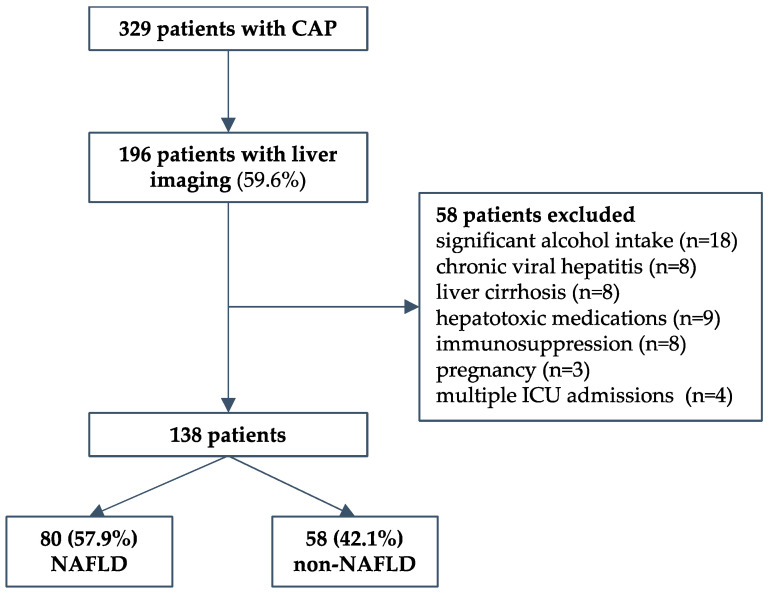
Study design flow chart.

**Figure 2 life-13-00036-f002:**
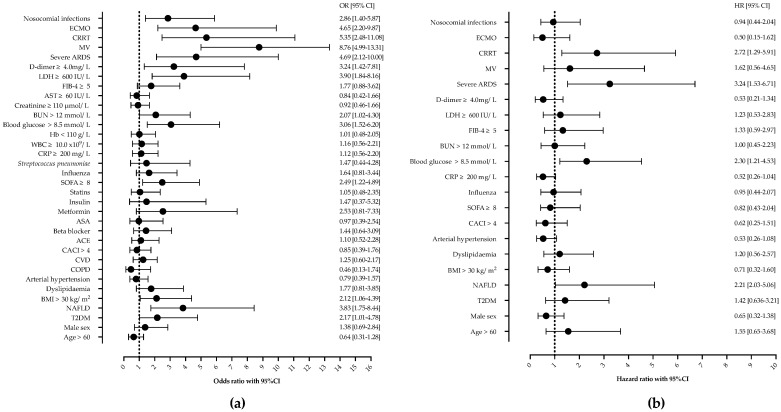
Univariable (**a**) and Cox proportional regression analysis (**b**) of factors associated with mortality in patients with severe community-acquired pneumonia. Shown are odds ratios (OR) and hazard ratios (HR) with corresponding 95% confidence intervals (95% CI). Abbreviations: NAFLD—nonalcoholic fatty liver disease, ARDS—acute respiratory distress syndrome, MV—mechanical ventilation, ECMO—extracorporeal membrane oxygenation, CRRT—continuous renal replacement therapy, LDH—lactate dehydrogenase, BUN—blood urea nitrogen, WBC—white-blood-cell count, ASA—acetylsalicylic acid, CACI—Charlson comorbidity index, CVD—cardiovascular disease.

**Figure 3 life-13-00036-f003:**
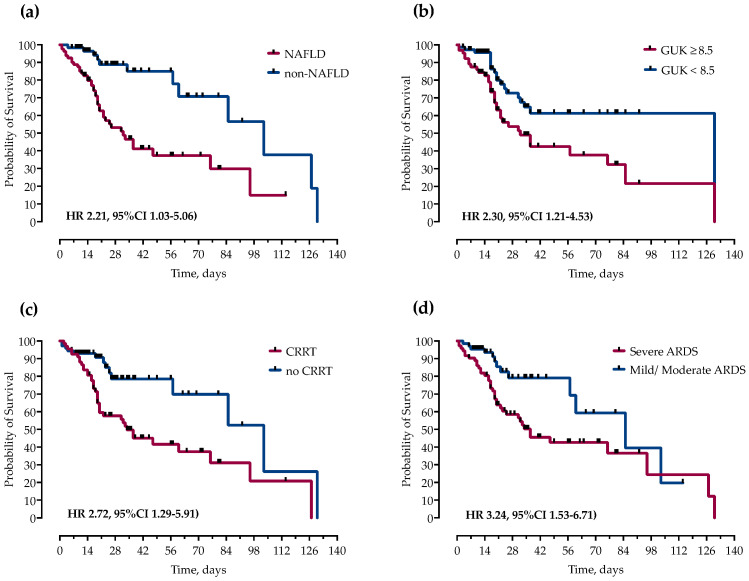
Kaplan–Meier curves and Cox proportional hazard ratios (HR) with corresponding 95% confidence intervals (95% CI) for probability of survival stratified by presence of (**a**) NAFLD, (**b**) ICU admission blood glucose levels, (**c**) continuous renal replacement therapy (CRRT) and (**d**) severity of ARDS.

**Figure 4 life-13-00036-f004:**
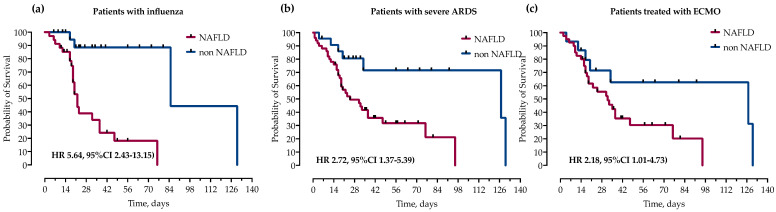
Kaplan–Meier curves and calculated hazard ratios (Mantel–Haenszel) (HR) with corresponding 95% confidence intervals (95% CI) for probability of survival stratified by the presence of NAFLD in subgroups of patients with (**a**) influenza, (**b**) severe ARDS (paO2/FiO2 < 100) and (**c**) treatment with ECMO. Statistical significance was calculated with a log-rank test.

**Figure 5 life-13-00036-f005:**
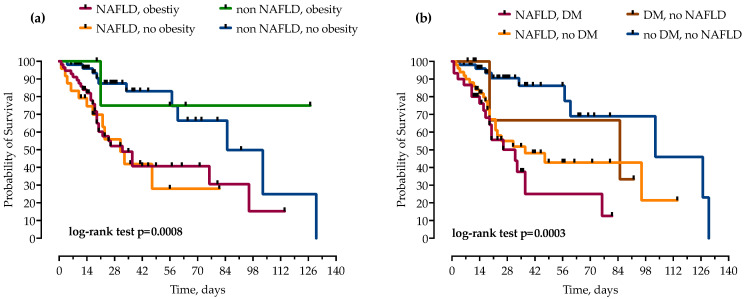
Using the Kaplan–Meier method, patients with NAFLD were stratified by concomitant presence of obesity (**a**) and diabetes mellitus (DM) (**b**), and impact on mortality was analyzed. Statistical significance was calculated with log-rank test; *p* < 0.05 was considered significant.

**Table 1 life-13-00036-t001:** Baseline patients’ characteristics *.

	NAFLD (n = 80)	Non-NAFLD (n = 58)	*p*-Value ^a^
Age, years, median (IQR)	61 (51–70)	60 (43–67)	0.3216
Male sex, n (%)	60 (75.00%)	35 (60.34%)	0.0933
Charlson age–comorbidity index	3 (1–4)	2 (1–3)	0.0929
Obesity	56 (70.00%)	5 (8.62%)	<0.0001
Body mass index, kg/m^2^	31 (28–34)	26 (25–28)	<0.0001
Smoker	22 (27.50%)	16 (27.59%)	>0.9999
**Comorbidities, n (%)**			
Diabetes mellitus	30 (37.50%)	7 (12.07%)	0.0009
Arterial hypertension	49 (61.25%)	25 (43.10%)	0.0394
Dyslipidemia	22 (27.50%)	9 (15.52%)	0.1039
Cardiovascular diseases	25 (31.25%)	11 (18.97%))	0.1195
Peripheral vascular disease	5 (6.25%)	3 (5.17%)	>0.9999
Chronic kidney disease	3 (3.75%)	1 (1.72%)	0.6388
Chronic obstructive pulmonary disease	6 (7.50%)	7 (12.07%)	0.3903
Neurological diseases	8 (10.00%)	8 (13.79%)	0.5927
**Chronic medications, n (%)**			
ACE inhibitors	31 (38.75%)	16 (27.59%)	0.2045
Other antihypertensive drugs	31 (38.75%)	19 (32.76%)	0.5907
Statins	22 (27.5%)	8 (13.79%)	0.0620
Metformin	15 (18.75%)	4 (6.90%)	0.0498
Another perioral anti-diabetic	12 (15.00%)	3 (5.17%)	0.0958
Insulin	10 (12.50%)	0 (0.00%)	0.0052
Antiplatelet agent	14 (17.50%)	7 (12.07%)	0.4745
**Duration of illness at admission**			
At hospital admission	5 (3–7)	4 (2.8–7)	0.2162
At ICU admission	6 (4–8)	5 (3–8)	0.4719

* Data are presented as frequencies (%) or medians with IQR. ^a^ Fisher exact or Mann–Whitney U test, as appropriate. Abbreviations: ACE inhibitors—Angiotensin-converting enzyme inhibitors, ICU—intensive care unit.

**Table 2 life-13-00036-t002:** Laboratory findings on admission in the ICU.

	NAFLD (n = 80)	Non-NAFLD (n = 58)	*p*-Value ^a^
C-reactive protein, mg/L	226 (142–370)	194 (133–267)	0.0971
Procalcitonin, µg/L	2.8 (0.55–18)	2.7 (0.42–17)	0.9797
Lactate, mmol/L	1.8 (1.3–3)	1.9 (1.4–2.9)	0.702
White blood cells, ×10^9^/L	10 (6.1–17)	11 (5–16)	0.681
Neutrophils–lymphocytes ratio	14 (7.7–28)	13 (6.6–23)	0.6051
Hemoglobin, g/L	118 (100–138)	125 (108–137)	0.4093
Platelets, ×10^9^/L	184 (113–256)	164 (102–229)	0.4267
Fibrinogen, g/L	5.8 (4.9–6.7)	5.5 (4.4–6.4)	0.6609
International normalized ratio (INR)	1.1 (1–1.3)	1.2 (1–1.3)	0.7171
D-dimer, mg/L	4.2 (2–4.3)	2.5 (0.94–4.3)	0.0816
Blood urea nitrogen, mmol/L	11 (6.1–17)	8.3 (5.8–13)	0.054
Creatinine, μmol/L	120 (84–204)	106 (87–153)	0.5098
Total bilirubin, μmol/L	14 (10–20)	12 (8.8–18)	0.1836
Aspartate aminotransferase, IU/L	73 (38–158)	45 (28–97)	**0.0067**
Alanine aminotransferase, IU/L	43 (22–81)	26 (16–53)	**0.0241**
Gamma-glutamyl transferase, IU/L	69 (42–152)	44 (24–93)	**0.0061**
Alkaline phosphatase, IU/L	84 (69–113)	76 (60–99)	0.0871
Lactate dehydrogenase, IU/L	497 (311–892)	305 (208–615)	**0.0019**
Serum albumins, g/L	28 (24–32)	29 (25–33)	0.5774
APRI score	1.1 (0.46–3.1)	0.71 (0.27–1.9)	0.0698
FIB-4 score	4.7 (2.1–9.3)	3.1 (1.3–7)	**0.0482**

Data are presented as medians with IQR; ^a^ Mann–Whitney U test.

**Table 3 life-13-00036-t003:** Etiology of pneumonia.

	NAFLD (n = 80)	Non-NAFLD (n = 58)	*p*-Value ^a^
Influenza	34 (42.50%)	22 (37.93%)	0.6036
*Streptococcus pneumoniae*	10 (12.50%)	8 (13.79%)	>0.9999
*Legionella pneumoniae*	4 (5.0%)	3 (5.17%)	>0.9999
Other bacteria ^b^	3 (3.75%)	4 (6.89%)	0.4535
Etiologically negative	29 (36.25%)	21 (36.21%)	>0.9999

Data are presented as frequencies (%); ^a^ Fisher exact test; ^b^
*S. aureus* (n = 2), *S. pyogenes* (n = 2), *H. influenzae* (n = 1), *Kl. pneumoniae* (n = 2).

**Table 4 life-13-00036-t004:** Clinical course and outcomes of community-acquired pneumonia.

	NAFLD (n = 80)	Non-NAFLD (n = 58)	*p*-Value ^a^
**Disease severity and modes of treatment**			
ARDS	55 (68.75%)	25 (43.10%)	0.0031
Mild or moderate ARDS ^b^	6 (7.50%)	3 (5.17%)	0.0078
Severe ARDS ^b^	49 (61.25%)	22 (37.93%)	
Invasive mechanical ventilation (IMV)	69 (86.25%)	37 (63.79%)	0.0038
Duration of IMV, days	13 (5–21)	8 (2–16)	0.0268
ECMO	40 (50.00%)	14 (24.14%)	0.0210
Duration of ECMO, days	9 (5–16)	7 (5–19)	0.2905
CRRT	50 (62.50%)	17 (29.31%)	0.0001
Duration of CRRT, days	10 (5–20)	12 (9–2)	0.8464
**Complications during hospitalizations**	67 (83.75%)	45 (77.59%)	0.3849
*Clostridiodes difficile* enterocolitis	3 (3.75%)	3 (5.17%)	0.6957
Hospital-acquired infections	42 (52.50%)	29 (50.00%)	0.8633
Ventilator-associated pneumonia	14 (17.50%)	10 (17.24%)	>0.9999
Hospital-acquired sepsis	27 (33.75%)	10 (17.24%)	0.0339
Acute kidney injury	58 (72.50%)	23 (39.66%)	0.0002
Acute heart failure	23 (28.75%)	12 (20.69%)	0.3255
Pneumothorax	6 (7.50%)	2 (3.45%)	0.4675
Empyema	2 (2.50%)	1 (1.72%)	>0.9999
Cardial arrest with return of spontaneous circulation	11 (13.75%)	4 (6.90%)	0.2713
**Outcomes**			
Death during hospitalization	40 (50.00%)	12 (20.69%)	0.0006
Time to death from hospital admission	18 (9–25)	46 (18–99)	0.0095
7-day mortality	9 (11.25%)	1 (1.72%)	0.0446
14-day mortality	16 (20.0%)	2 (3.44%)	0.0043
28-day mortality	32 (40.0%)	5 (8.6%)	<0.0001
90-day mortality	39 (48.7%)	9 (15.5%)	<0.0001

Data are presented as frequencies (%) or medians (IQR), ^a^ Fisher exact or Mann–Whitney U test, as appropriate. ^b^ according to the Berlin definition [35], Abbreviations: NAFLD—nonalcoholic fatty liver disease, ARDS—acute respiratory distress syndrome, IMV—invasive mechanical ventilation, ECMO—extracorporeal membrane oxygenation, CRRT—continuous renal replacement therapy.

## Data Availability

The datasets generated during and/or analyzed during the current study are available from the corresponding author on reasonable request.

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
