# Peer review of "The Impact of Nonalcoholic Fatty Liver Disease on Severe Community-Acquired Pneumonia Outcomes"

_life, 2022, doi:10.3390/life13010036_

Round 1
Reviewer 1 Report
Journal: Life
Manuscript ID: life-2068478
Title: The impact of nonalcoholic fatty liver disease on severe community acquired pneumonia outcomes
Authors: Branimir Gjurašin et al.
The authors of this retrospective study tried to explore the possible impact of non-alcoholic fatty liver disease on the severity, and outcomes, including mortality, of patients with severe community-acquired pneumonia admitted to the intensive care unit. The following points should be considered:
Major Comments
1. I would suggest using type “2” diabetes mellitus instead of type “II” diabetes mellitus.
2. According to the authors, 4 patients were excluded from the study due to multiple episodes. However, it is not clear what the multiple episodes refer to. Moreover, it would be helpful if the authors presented all the predefined exclusion criteria of the present study. In addition, only patients with abdominal ultrasonography were eligible? For example, were patients with abdominal computer tomography or other liver imaging techniques excluded?
3. In addition, why an abdominal ultrasound was performed on these patients with severe community-acquired pneumonia, and when (e.g., at admission, during hospitalization)? Did the authors consider the potential presence of features of liver steatosis due to sepsis?
4. For completeness, briefly define APRI, FIB-4, and SOFA scores and how the authors calculated these scores. Moreover, please add the relevant references.
5. It would be helpful if the authors could clarify how they excluded other causes of liver steatosis and coexisting liver diseases.
6. Please add the reference regarding the Berlin criteria used for the ARDS severity definition.
7. Did the authors consider performing multivariable logistic regression analyses for all the prespecified outcomes of the study?
8. Did the authors calculate the length of stay of the patients?
9. Did the authors perform univariable analysis for all the variables examined in the present study?
10. According to the authors, “Age, gender, obesity and other comorbidities were not associated with mortality in our model.” Do the authors mean the multivariable Cox proportional hazards regression model? If so, please update Table 5 by adding all the variables included in the model. Did the authors consider including also hypertension, metformin, and insulin treatment?
11. Please define all the abbreviations used in the tables and figures of the present manuscript.
12. Another study limitation is the lack of data regarding NAFLD staging and insulin resistance.
Author Response
Reviewer 1
The authors of this retrospective study tried to explore the possible impact of non-alcoholic fatty liver disease on the severity, and outcomes, including mortality, of patients with severe community-acquired pneumonia admitted to the intensive care unit. The following points should be considered:
Major Comments
- I would suggest using type “2” diabetes mellitus instead of type “II” diabetes mellitus.
ANSWER: We changed “type II” in “type 2” DM in the manuscript.
- According to the authors, 4 patients were excluded from the study due to multiple episodes. However, it is not clear what the multiple episodes refer to. Moreover, it would be helpful if the authors presented all the predefined exclusion criteria of the present study. In addition, only patients with abdominal ultrasonography were eligible? For example, were patients with abdominal computer tomography or other liver imaging techniques excluded?
ANSWER: We thank the reviewer for this comment. In revised version of the manuscript, we added the exclusion criteria.
“Predefined exclusion criteria were: immunosuppression; malignancies; autoimmune diseases; pregnancy; HIV; chronic viral hepatitis; presence of other chronic liver disease (hemochromatosis, Wilson's disease, toxic hepatitis, deficiency of alpha-1-antitrypsin, liver autoimmune disease); consumption of alcohol > 20 g/day.”
Regarding multiple episodes, it refers to more than one ICU admission during the same hospitalization. We corrected this statement and changed the study flow-chart.
Regarding NAFLD definition, we included only patients who had abdominal US during hospitalization. We did not consider liver steatosis on CT scans for several reasons: the majority of patients had CT pulmonary angiography which has limited sensitivity in detecting steatosis, especially in the context of severe inflammation (as compared to US). The presence of liver steatosis on CT is usually defined as mean hepatic CT attenuation < 40 HU or mean CT hepatic attenuation minus mean splenic CT attenuation of less than 10 HU. However, in the setting of sepsis this approach usually does not detect mild/moderate steatosis. Next, only few patients who did not have US, had CT scan. We have added following paragraph:
“The liver steatosis was assessed by ultrasound in all included patients by experienced radiologist during hospitalization, and defined as finding of liver parenchyma with in-creased echogenicity and sound attenuation, as compared to kidney and spleen echogenicity”
- In addition, why an abdominal ultrasound was performed on these patients with severe community-acquired pneumonia, and when (e.g., at admission, during hospitalization)? Did the authors consider the potential presence of features of liver steatosis due to sepsis?
ANSWER: Abdominal ultrasound was performed at physician discretion. Since this was a retrospective study and there is no protocol or official guidelines when to perform abdominal US in patients with CAP, not all patients had US performed. However, a large proportion of hospitalized patients with CAP underwent US (56%, as shown in Figure 1) due to the ICU good-clinical practice (e.g. evaluation for empyema or metastatic infection). Indeed, one could argue that patients with elevated liver enzymes more frequently had US, and this could lead to “confounding by indication” bias. This is now added in limitations:
“…not all hospitalized patients were routinely screened for liver steatosis and since there were no protocol for US in ICU patients with CAP this could lead to confounding by indication bias;…”
As we mentioned above, liver steatosis was defined as increased echogenicity and sound attenuation by experienced radiologist. Except in liver steatosis, the similar could be seen in severe hypoxic hepatitis, however as part of US liver protocol, liver doppler was performed and experienced radiologist could distinguish between these conditions. We believe that finding of fatty liver could not be due to the sepsis. Indeed, more objective methods are needed, such as fibroelastography with continuous attenuation parameter that is not routinely performed.
- For completeness, briefly define APRI, FIB-4, and SOFA scores and how the authors calculated these scores. Moreover, please add the relevant references.
ANSWER: We added the relevant references for APRI, FIB-4 and SOFA scores.
- It would be helpful if the authors could clarify how they excluded other causes of liver steatosis and coexisting liver diseases.
ANSWER: We thank the reviewer for that comment. We have clarified it in the revised manuscript. Competing causes of liver steatosis and coexisting liver diseases were based on previously described exclusion criteria collected from patients’ medical charts. As per ICU protocol, all patients were tested for HIV, HBsAg and anti-HCV on ICU admission.
- Please add the reference regarding the Berlin criteria used for the ARDS severity definition.
ANSWER: The Berlin criteria used for ARDS severity was cited.
- Did the authors consider performing multivariable logistic regression analyses for all the prespecified outcomes of the study?
ANSWER: We thank the reviewer for this suggestion. We performed multivariable logistic regression analysis for other outcomes, and as you suggested we have now included it in the study results. Originally, we worried that including this data would make the manuscript harder to read and would not provide additional data, so we focused on the survival analysis.
“…Next, we performed a series of multivariable logistic regression analyses to identify risk factors associated with previously predefined outcomes of CAP. The need for invasive mechanical ventilation was associated with BUN ≥ 12 mmol/L (OR 2.31, 95%CI 1.08-6.09), LDH ≥ 600 IU/L (OR 22.88, 95%CI 2.89-89.8) and NAFLD (OR 3.31, 95%CI 1.37-8.06). Factors associated with severe ARDS were arterial hypertension (OR 2.66, 95%CI 1.17-6.06), NAFLD (OR 5.24, 95% 2.32-11.08) and influenza (OR 2.92, 95%CI 1.28-6.64). The need for ECMO was associated with FIB-4 score ≥ 5 (OR 2.5, 95% CI 1.19-5.23), admission glucose ≥ 8.5mmol/L (OR 3.43, 95%CI 1.55-7.60) and severe ARDS (OR 2.43, 95%CI 1.12-5.27%). Independently associated with CRRT was only ECMO treatment (OR 41, 95%CI 12.99-130.91%)., Nosocomial infections were associated with ECMO treatment (OR 4.81, 95%CI 2.21-10.44) and male sex (OR 2.53, 95%CI 1.13-5.69) in our analysis.”
- Did the authors calculate the length of stay of the patients?
ANSWER: Yes, and the duration of hospitalization is shown in the manuscript:
“The duration of hospitalization was similar between groups (19 [14-35] vs 24 [14-42], p = 0.1277).”
- Did the authors perform univariable analysis for all the variables examined in the present study?
ANSWER: Yes. We performed univariable analysis for all the variables examined in the present study and originally showed only those with statistical significance in Table 5. This is now more clearly presented; we have built Figure 2 (panel a) with forest plot including all variables that were tested.
- According to the authors, “Age, gender, obesity and other comorbidities were not associated with mortality in our model.” Do the authors mean the multivariable Cox proportional hazards regression model? If so, please update Table 5 by adding all the variables included in the model. Did the authors consider including also hypertension, metformin, and insulin treatment?
ANSWER: Yes, presented this data more clearly in Figure 2.
These variables were selected due to the biological plausibility. We did not include metformin, statins or antihypertensive drugs in our model since they were not associated with mortality in univariable analysis.
If the reviewer agrees, we think it is unnecessary to build additional table with the data on Cox analysis which would include nonsignificant variables.
- Please define all the abbreviations used in the tables and figures of the present manuscript.
ANSWER: In the revised version we now defined all the abbreviations in the tables and figures.
- Another study limitation is the lack of data regarding NAFLD staging and insulin resistance.
ANSWER: We agree with the reviewer, and we added insulin resistance and NAFLD staging in study limitations.
Reviewer 2 Report
Dear Editor,
I have read with great interest the manuscript submitted by Gjurašin et al. entitled “The impact of nonalcoholic fatty liver disease on severe community acquired pneumonia outcomes” to Life. However, there are some issues that need to be addressed before further processing.
Introduction
I believe that the concept of MAFLD should be reported in the introduction, because the message the authors are trying to report is that worse metabolic status is associated with worse outcomes of respiratory infection as it happened with COVID-19. Also, it is mandatory to report some data on mortality/ICU admission in patients with T2DM (often associated with NAFLD) or other metabolic comorbidities to make the reader understand with the aim of the study is important.
Materials and Methods
The exclusion criteria are not clear in the text, please revise accordingly.
What test was used to check the parametric distribution of data?
How was sample size calculated?
How were the multivariate models chosen?
If the authors have data of waist circumference, I suggest using the fatty liver index (FLI) to quantify steatosis. Was steatosis severity evaluated by ultrasound (e.g. Hamaguchi score?)? This can really help the analysis and patient stratification.
Author Response
Reviewer 2
Dear Editor,
I have read with great interest the manuscript submitted by Gjurašin et al. entitled “The impact of nonalcoholic fatty liver disease on severe community acquired pneumonia outcomes” to Life. However, there are some issues that need to be addressed before further processing.
- Introduction
I believe that the concept of MAFLD should be reported in the introduction, because the message the authors are trying to report is that worse metabolic status is associated with worse outcomes of respiratory infection as it happened with COVID-19. Also, it is mandatory to report some data on mortality/ICU admission in patients with T2DM (often associated with NAFLD) or other metabolic comorbidities to make the reader understand with the aim of the study is important.
ANSWER: We thank the reviewer for this comment. We have added following paragraph in Introduction:
“Metabolic syndrome, defined by a combination of abdominal obesity, dyslipidemia, arterial hypertension or insulin resistance, is becoming increasingly prevalent, with more than 50% increase in less than a decade [3]. While each component of metabolic syndrome was reported to increase mortality of CAP, their combined impact may be even more pronounced. Patients with either insulin resistance or type 2 diabetes mellitus (T2DM) were shown to have 1.3- and 2.8- fold increase in CAP mortality, respectively [4]. Similarly, obese patients more frequently develop CAP and require ICU treatment [5,6]. In COVID-19, dyslipidemia was associated with increased disease severity and hospital mortality [7]. “
However, we decided not to introduce the term of MAFLD. We are aware of the ongoing debate and the new proposed definition of MAFLD (J Hepatol. 2020 Jul;73(1):202-209.), which requires presence of obesity and/or DM, and in patients with normal BMI at least two components of metabolic syndrome. However, this has not yet been included in official guidelines. Presently, the definition of NAFLD as reported in most guidelines and recent publications is based on the presence of steatosis in the absence of significant ongoing or recent alcohol consumption and other known causes of liver disease, and this is the definition of NAFLD that we used in our study. We believe that the uniform use of the term “NAFLD” contributes to the comprehensives of our study.
- Materials and Methods
The exclusion criteria are not clear in the text, please revise accordingly.
ANSWER: We thank the reviewer for this comment. In revised version of the manuscript, we added the exclusion criteria.
“Predefined exclusion criteria were: immunosuppression; malignancies; autoimmune diseases; pregnancy; HIV; chronic viral hepatitis; presence of other chronic liver disease (hemochromatosis, Wilson's disease, toxic hepatitis, deficiency of alpha-1-antitrypsin, liver autoimmune disease); consumption of alcohol > 20 g/day.”
Regarding multiple episodes, it refers to more than one ICU admission during the same hospitalization. We have corrected this statement and changed the study flow-chart.
- What test was used to check the parametric distribution of data?
ANSWER: Shapiro–Wilk test was used as normality test. This is now added in methods. “Shapiro–Wilk test was used to check if a continuous variable follows a normal distribution.”
- How was sample size calculated?
ANSWER: This was a retrospective cohort study that included all consequently hospitalized patients with predefined inclusion criteria, so the sample size and power was not calculated. A post hoc analysis showed power of 88.5% for mortality outcome, but since this is not informative and could be misleading, we decided not to include it in the manuscript.
- How were the multivariate models chosen?
ANSWER: We included variables that were associated with mortality on univariable analysis and those with biological plausibility, and we have added the description in the text and bult new figure present the data on univariable and multivariable analysis (Figure 2).
- If the authors have data of waist circumference, I suggest using the fatty liver index (FLI) to quantify steatosis. Was steatosis severity evaluated by ultrasound (e.g. Hamaguchi score?)? This can really help the analysis and patient stratification.
ANSWER: Unfortunately, due to the retrospective design of the study we were not able to collect data on waist circumference (abdominal obesity) as reviewer suggested. Data on steatosis grade was not available, as is now reported in the study limitations.
Round 2
Reviewer 1 Report
Journal: Life; ID: life-2068478 (Revised)
Authors: Branimir Gjurašin et al.
Manuscript Title: The impact of nonalcoholic fatty liver disease on severe community acquired pneumonia outcomes
The authors have tried to address my comments and suggestions and revised their manuscript accordingly. The revised manuscript is an interesting paper focusing on an important scientific topic given the current pandemic. However, the study design and subsequent limitations should be considered with caution when interpreting the findings. No further comments.
Reviewer 2 Report
The authors have edited the manuscript according to reviewers suggestions. The manuscript can now be accepted for publication.